

# Study on mid-term outcomes of atherectomy for patients with femoral popliteal artery lesions with different Global Limb Anatomic Staging System grades

Yanyu Yue[1], Youjia Zhang[2], Liang Zhang[3], Zheng Gao[2], Xiaolong Du[2] and Feng Ran[1]

[1] Department of Vascular Surgery, Nanjing Drum Tower Hospital Clinical College of Nanjing University of Chinese Medicine, Nanjing, Jiangsu, China

[2] Department of Vascular Surgery, Nanjing Drum Tower Hospital, Affiliated Hospital of Medical School, Nanjing University, Nanjing, Jiangsu, China

[3] Department of Radiology, Nanjing Drum Tower Hospital, Affiliated Hospital of Medical School, Nanjing University, Nanjing, Jiangsu, China

## ABSTRACT

**Objective**. To investigate the mid-term efficacy and patency rate of TurboHawk peripheral plaque excision system in the treatment of femoral popliteal artery lesions with different Global Limb Anatomic Staging System (GLASS) grades.

**Methods**. The clinical data of 141 patients with femoral popliteal arteriosclerosis obliterans who were treated with TurboHawk from January 2018 to July 2022 in our institution were retrospectively analyzed. There were 109 male patients and 32 female patients. Recordings were made of the patient's symptoms of limb ischemia, technical success rate, primary patency rate of target vessels, ankle brachial index (ABI), GLASS grades, postoperative complications, and a statistical analysis with the patient's preoperative treatment was conducted.

**Results**. All patients had improved limb ischemia symptoms to varying degrees after surgery, with a technical success rate of 100% (femoral artery puncture and superficial femoral artery recanalization) without bleeding, hematoma, pseudoaneurysm, arteriovenous fistula or other complications. The follow-up period was 1-24 months, during which the severity of claudication, resting pain, and toe ulcers significantly improved. The primary patency rate of the target vessel was 98.58% (139/141), and the ABI significantly increased on the second day, three months, and six months after surgery compared to before surgery. No major adverse events were found during follow-up. The patency rates at 1, 6, 12 and 24 months after intervention were 100%, 80%, 75% and 60% respectively.

**Conclusion**. The mid-term efficacy and patency rate of TurboHawk in the treatment of femoral popliteal artery lesions with GLASS I patients have the best mid-term prognosis, the highest mid-term survival rate, and the highest vascular patency. The plaque removal system has proven to be an effective treatment for individual localized chronic total occlusion lesions. Additionally, the TurboHawk system provides a safe and minimally invasive treatment alternative for superficial femoral artery conditions, achieving significant therapeutic results within a brief period.

Corresponding authors
Xiaolong Du, dd0341@163.com
Feng Ran, doctor_ran@163.com

## INTRODUCTION

Arteriosclerosis obliterans (ASO) of the lower extremities is a common condition in vascular surgery, characterized by arterial stenosis or occlusion due to atherosclerosis. As the disease progresses, it can lead to severe limb ischemia, potentially resulting in death or significant morbidity (*Akagi et al., 2018*). Currently, percutaneous balloon dilation and stent placement are the standard treatments (*Hiramoto et al., 2018*). However, managing complex lesions in the femoral and popliteal arteries remains a significant clinical challenge, especially in cases of extended segmental occlusions or multiple stenotic sites in the superficial femoral artery (SFA) (*Van der Vijver-Coppen et al., 2018*). For decades, vascular surgery has grappled with these issues. While short occlusions or stenoses can be treated with percutaneous transluminal angioplasty (PTA), the long-term patency rates for occlusive lesions ranging from 4 to 10 cm have been disappointing (*Vossen et al., 2018*). Even with the addition of stenting to angioplasty, outcomes have not improved significantly, a situation that persists despite careful monitoring and prompt reintervention (*Kansal et al., 2019*). The TurboHawk peripheral plaque excision system introduces a new technique by excising atherosclerotic plaques, which helps alleviate arterial stenosis and improve limb perfusion (*Adams & Patel, 0000*). This retrospective analysis evaluates the efficacy of the system in treating femoropopliteal lesions across various Global Limb Anatomic Staging System (GLASS) classifications. The following report details these findings.

## MATERIAL AND METHODS

### Patients' recruitment

A retrospective evaluation was conducted to assess the therapeutic efficacy and practicability of the TurboHawk peripheral plaque excision system for patients with superficial femoral atherosclerosis (SFA). The data for patients treated for femoral popliteal arteriosclerosis obliterans between January 2018 and July 2022 in our institution were retrieved and analyzed. Inclusion criteria included (1) verified occlusion or stenosis of the femoral popliteal artery *via* Computed Tomography Angiography (CTA) or angiography, (2) underwent TurboHwak treatment, (3) patients with complete clinical data, (4) having signed the informed consent form. Exclusion criteria included (1) allergic to contrast agents, (2) severe cardiopulmonary, hepatic, renal, or coagulation disorders, and other contraindications to endovascular procedures, (3) patients with pure venous ulcers, pure traumatic wounds, acute limb ischemia, embolic disease, and nonatherosclerotic chronic vascular conditions of the lower extremity. Written informed forms were obtained from participants. The study protocol was approved by institutional review board of the Affiliated Hospital of Nanjing University Medical School (No:2018-015-05).

## Atherectomy procedure

During operation, the TurboHawk is advanced to the stenosis. Activating the motor retracts the positioning rod, angling the catheter tip and pressing the windowed rotary blade against the vessel wall. As the device traverses the narrowed segment, the blade cuts through the plaque, collecting material into the catheter's conical tip. The accumulated material is then cleared from the catheter. A final angiography verifies successful vascular recanalization. Stents are generally avoided, used only for flow-limiting dissection or significant residual stenosis after plaque excision and angioplasty.

Technical success was defined by angiographic confirmation of vascular lumen stenosis reduction to $\leq 30\%$, absence of flow-compromising arterial dissection, and avoidance of target vessel perforation or embolization. Authentic lumen identification relied on criteria such as continuous linear contrast agent flow during angiography, absence of stagnation, collateral vessel imaging, and guidewire proximity to the intima as verified by endovascular ultrasound.

## Postoperative management and follow-up

Postoperative management included rivaroxaban for anticoagulation and cilostazol for preventing intimal hyperplasia, with a transition to long-term maintenance dosages as appropriate. Statin therapy was advised unless contraindicated. Patients were monitored at intervals of 1, 6, 12 and 24 months following the atherectomy procedure, in accordance with the study's protocol. During each visit, patients underwent comprehensive clinical reevaluations, which included interviews, physical inspections, and objective hemodynamic tests. The ankle-brachial index (ABI) and/or duplex ultrasound imaging was utilized to evaluate the patency of the treated vessels. Vascular surgeons reviewed patients between scheduled follow-ups. Immediate attention was given if target lesion revascularization or amputation was required. Primary patency, defined as less than 50% restenosis confirmed by duplex ultrasound without further interventions, was closely monitored.

## Statistical analysis

Statistical assessments utilized IBM SPSS Statistics software, version 20.0 (IBM Corp., Armonk, NY, USA). Quantitative data were described as mean $\pm$ standard deviation (SD). Paired t-tests were employed to discern statistical discrepancies between groups. Categorical variables were presented as frequencies and percentages. A $p$-value of less than 0.05 was considered statistically significant. The Kaplan–Meier method estimated cumulative patency and limb salvage rates over the follow-up period.

## RESULTS

### Participant characteristics

A total of 141 patients (109 males, 32 females) were recruited for this study. The average age was $71.35 \pm 0.72$ years for males and $78.32 \pm 1.42$ years for females. Patients had experienced limb ischemia for 1 to 3 years, presenting with decreased lower limb skin temperature, pallor, cyanosis, intermittent claudication (33 cases), resting pain (37 cases), distal mild ischemic ulcers (33 cases), and tissue ischemic necrosis with gangrene (38

**Table 1  Baseline characteristics of patients.**

| Variable | $n = 141$ |
| --- | --- |
| Age (years) | Male (71.35 ± 0.72) Female (78.32 ± 1.42) |
| Sex (%) | Male 109 (77.3) Female 32 (22.7) |
| Hpertension | 33 (23.4) |
| Diabetes mellitus | 72 (51.1) |
| Myocardial infarction | 39 (27.7) |
| Cerebral infarction | 21 (14.9) |
| Hyperlipidemia (triglycerides) | 36 (25.5) |
| Limp | 120 (85.1) |
| Pain | 85 (60.3) |
| Gangrene | 42 (29.8) |
| Smoking | 37 (26.2) |
| Tumour | 37 (26.2) |
| Renal insufficiency | 52 (36.9) |

cases). All patients were classified between Rutherford stages 3 and 6. The GLASS stage distribution was: GLASS I in 33 patients (23%), GLASS II in 37 (26%), GLASS III in 33 (23%), and GLASS IV in 38 (27%) (Table 1).

## Clinical outcomes

The entire cohort of patients successfully underwent surgery, achieving complete revascularization of the targeted vessels. Technical success was recorded in 135 cases (95.74%), with six instances of significant dissection necessitating the placement of Everflex self-expanding bare metal stents. Five patients experienced critical arterial occlusions leading to ruptures during high-pressure balloon angioplasty; these were managed with compression bandaging, which effectively restored blood flow. Post-intervention outcomes indicated significant improvement in 65 patients (46%) and moderate improvement in 76 patients (54%), as evidenced by reduced lumen stenosis and significant enhancement of the ABI (Fig. 1). In addition, we analyzed the ABI index in different afflicted limbs. Our data showed that there is no statistical significance on ABI index before surgery between afflicted limbs (Fig. 2A). There were also no statistically significant differences in the proportion of affected limbs among the different groups (Fig. 2B). However, the differences in pre-procedural claudication distances among the groups were statistically significant, with the GLASS I group demonstrating a notably higher claudication distance compared to the other groups (Fig. 2C).

## Follow up outcomes

Patients were monitored postoperatively for periods ranging from 1 to 24 months. 116 out of 141 patients were monitored during follow-up. Across the cohort, there was a significant improvement in lower limb ischemia symptoms. At the one-year follow-up, restenosis was observed in six patients within the initially treated segments. However, these cases did not require further surgical intervention due to the lack of clinical symptoms and were managed conservatively. After 24 months, five patients developed secondary stenosis accompanied
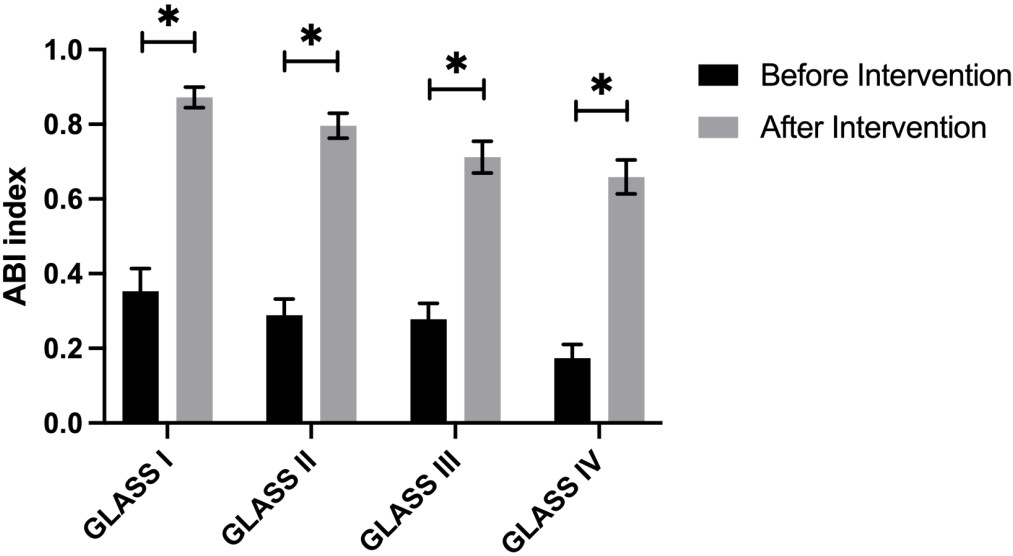

**Figure 1  Baseline characteristics of patients.**

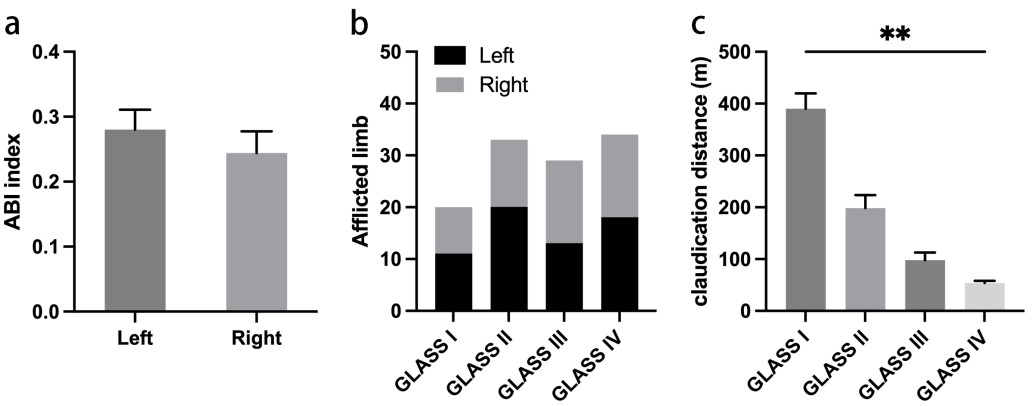

**Figure 2  Data on afflicted limbs before intervention.** (A) Pre-intervention ABI index on different afflicted limbs. (B) The proportion of afflicted limbs among different groups. (C) Claudication distance among different groups.

by symptoms that necessitated surgical reintervention, which effectively alleviated their symptoms. Figure 3 showed that ABI index gradually decreased during follow-up in all GLASS I-IV groups. Additionally, we analyzed the ABI index with survival curve (Fig. 4). The results showed that ABI degrees of freedom in all groups classified by GLASS I to IV were less than 0.6 during follow-up. Patients in group IV (GLASS IV) exhibit the highest proportion of individuals with ABI values less than 0.6, reflecting more severe vascular
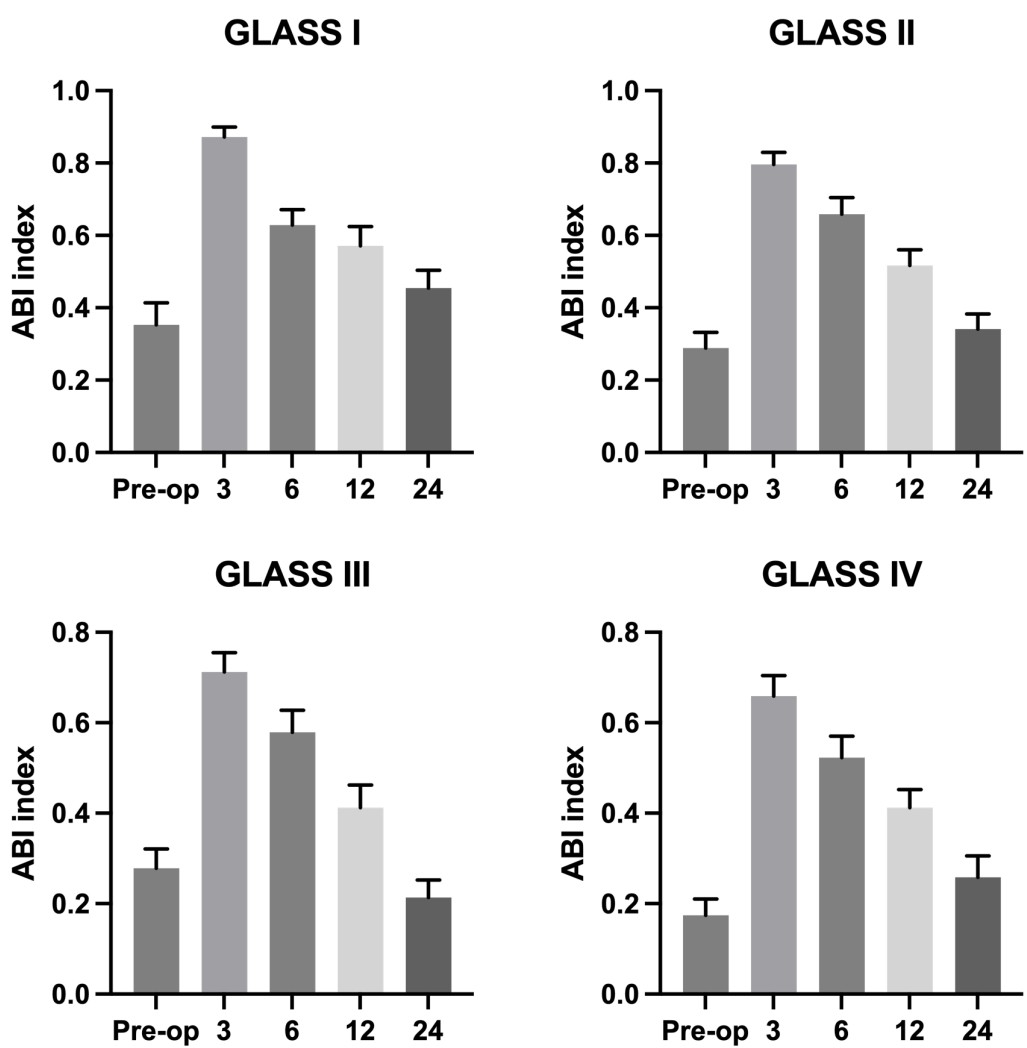

**Figure 3** ABN index of different groups during follow-up period.

impairment, which underscores the differential prognosis associated with each GLASS stage.

Patients without type 2 diabetes mellitus exhibited superior primary and secondary patency rates postoperatively compared to those with diabetes, consistent with existing literature that identifies diabetes as a negative prognostic factor in vascular intervention outcomes. Our findings support established correlations that identify lesion length as a critical determinant of long-term patency. Specifically, lesions with an average length of 7.5 cm, of which 28% exceeded 10 cm, showed improved patency following atherectomy and selective balloon dilation. Additionally, the GLASS classification proved to be a valuable prognostic tool, with higher stages being associated with increased rates of reintervention and restenosis.

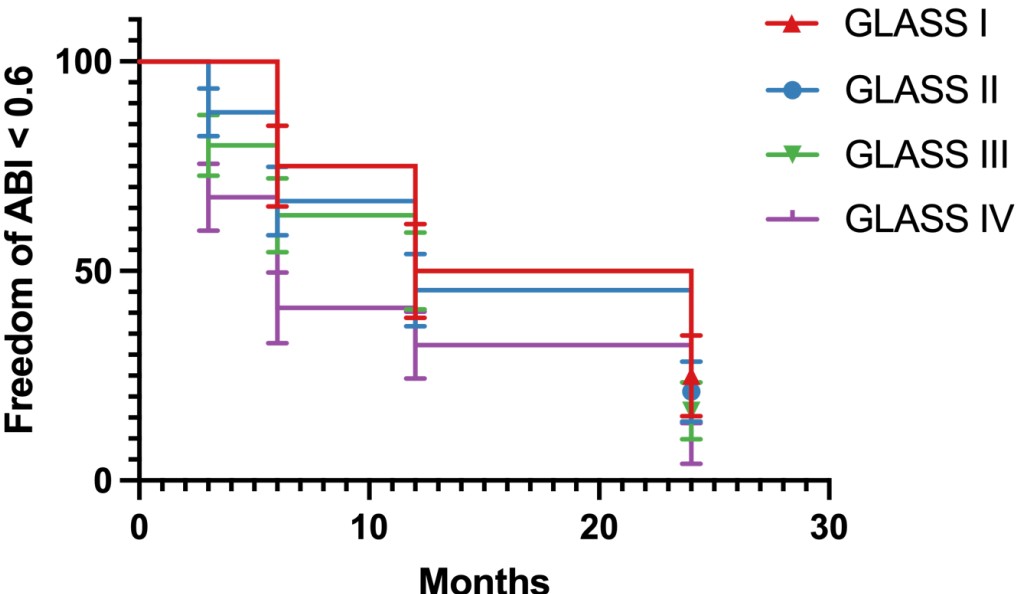

**Figure 4** Freedom of ABI value less than 0.6 among different groups during follow-up period.

# DISCUSSION

The incidence of arteriosclerosis obliterans (ASO) has been increasing annually, affecting over 200 million individuals globally (*Criqui & Aboyans, 2015*). ASO represents a significant public health concern. Current mainstream treatments, such as percutaneous balloon angioplasty and stent implantation, have notable limitations, especially for long-term management of lower limb ASO with severe stenosis or involvement of knee-adjacent femoropopliteal arteries. Atherectomy emerges as a minimally invasive alternative that mitigates complications associated with surgical bypass and reduces operation time. This technique not only alleviates ischemic symptoms but also reduces the risk of amputations, thereby enhancing quality of life for patients. Previous study has indicated that short to medium-length femoropopliteal lesions can be effectively managed with atherectomy, yielding favorable technical and 6-month clinical outcomes, particularly in primary lesions compared to restenoses (*Liang et al., 2021*).

Recent studies have highlighted the importance of ABI in patients with arteriosclerosis, identifying a reduction in ABI as an independent risk factor for arteriosclerosis obliterans and a predictor of both total and cardiovascular mortality. A decreased ABI is associated with a three- to six-fold increase in the mortality of vascular disease. With sensitivity and specificity rates of 95% and 99%, respectively, ABI is an ideal non-invasive indicator for early detection of lower limb peripheral arterial diseases (*Hasanadka et al., 2008a*; *Baxter & Polak, 1993*). Pre-procedure ABI values below 0.4 have been correlated with a higher risk of losing patency post-intervention (*Surowiec et al., 2005*). Furthermore, ABI has demonstrated high sensitivity and specificity for detecting lesions identified by angiography (*Carter, 1968*). The ABI values in diabetic patients, particularly those with

knee involvement, were closely monitored due to their higher risk of developing gangrene and refractory ulcers.

Our study targeted patients with Rutherford grades 3–6, who are often afflicted with long-term smoking, hypertension, type 2 diabetes mellitus, coronary heart disease, or hyperlipidemia. A substantial number of these patients demonstrated symptomatic improvement following treatment, thereby confirming the effectiveness of interventions for arterial atherosclerosis. Predominantly, the participants had a history of type 2 diabetes mellitus and smoking, factors that aggravate atherosclerosis and elevate the risk of amputation. We observed a significant enhancement in postoperative ABI values, notably in patients with GLASS Class I, which was associated with superior recovery outcomes and a diminished incidence of secondary restenosis in comparison to those in GLASS Classes III and IV. Importantly, patients with severe calcified stenosis, especially in areas of high biomechanical stress, derived benefit from atherectomy as an alternative to stent implantation (*Bontinis et al., 2023*; *Minko et al., 2011*). Additionally, our findings revealed, although without statistical significane, patients exhibiting poorer ABI values in the right leg compared to the left leg.

Our results also demonstrated that male patients experienced a significantly longer overall claudication distance than female patients, with males also displaying a higher average distance. Additionally, analysis of the lameness distance across different affected limbs revealed a longer lameness distance in the left leg. Postoperative findings indicate that ABI values on the healthy side of affected limbs consistently remained above 0.9, whereas ABI measurements on the healthy limb showed a minor decline post-surgery. Furthermore, our data suggests a progressive decrease in ABI value changes with increasing age, coupled with an intensification of blood vessel occlusion. Notably, older patients exhibited a less pronounced recovery in ABI following surgery, underscoring the impact of age on postoperative recovery. Moreover, in our experience, patients with three combined branches of the knee can have their daily needs met by opening one branch of the knee. After intravascular therapy in the femoral and popliteal artery segments, the hemodynamics and clinical success rate of patients with ABI $\geq 0.5$ or at least one tibial vessel with open blood flow were significantly improved (*Hasanadka et al., 2008b*; *Tokuda et al., 2020*; *Lawrence & Gloviczki, 2019*; *Gray et al., 2002*).

Technological advancements and increased procedural expertise have considerably improved the efficacy of endovascular interventions. Innovations such as the micropuncture system, specialized guide wires for chronic total occlusions, supporting catheters, and the true lumen system have facilitated the development of technology like dorsalis pedis planta artery arch and bidirectional subintimal angioplasty. These advancement have expanded the scope of traditional interventions. Specifically, the TurboHawk peripheral plaque excision system has been shown to significantly enhance lumen volume and diameter, rapidly restoring blood flow and improving patency rates shortly after surgery. However, evidence suggests that the initial benefits of the TurboHawk system may be tempered by a tendency toward restenosis, primarily due to intimal hyperplasia (*Bracale et al., 2016*; *Guan et al., 2018*). Drug-coated balloons (DCBs) offer an additional therapeutic benefit by facilitating direct contact of the drug with the

endothelium, thus inhibiting hyperplasia and mitigating inflammatory responses. This action substantially reduces the likelihood of restenosis in the treated vessel. Consequently, the integration of atherectomy with other interventional strategies has proven beneficial for patients with ASO.

## CONCLUSIONS

In general, our study demonstrated that patients with GLASS I who underwent plaque resection exhibited superior mid-term outcome. The plaque removal system has proven to be an effective treatment for individual localized chronic total occlusion lesions. Additionally, the TurboHawk system provides a safe and minimally invasive treatment alternative for superficial femoral artery conditions, achieving significant therapeutic results within a brief period. However, due to the retrospective nature of this study and the strigent inclusion criteria, further research involving a larger cohort and longer follow-up periods is essential to validate these findings and to fully evaluate the long-term efficacy of the interventions discussed.

### Funding

This work is supported by the Project funded by the National Natural Science Foundation of China (82100517), the Natural Science Foundation of Jiangsu Province (SBK2020042213), the fundings for Clinical Trials from the Affiliated Drum Tower Hospital, Medical School of Nanjing University, and the 2020 Jiangsu Province Shuangchuang Ph.D. Introducing Talent Project and Natural Science Foundation of Nanjing of Xiaolong Du. Xiaoqiang Li of the Department of Vascular Surgery, Nanjing Drum Tower Hospital helped with the review of our research on topics. The funders had no role in data collection and analysis, decision to publish, or preparation of the manuscript.

### Grant Disclosures

The following grant information was disclosed by the authors:
National Natural Science Foundation of China: 82100517.
Natural Science Foundation of Jiangsu Province: SBK2020042213.
The Affiliated Drum Tower Hospital, Medical School of Nanjing University.
The 2020 Jiangsu Province Shuangchuang Ph.D. Introducing Talent Project.
Natural Science Foundation of Nanjing of Xiaolong Du.

### Competing Interests

The authors declare there are no competing interests.

### Author Contributions

- Yanyu Yue conceived and designed the experiments, performed the experiments, analyzed the data, prepared figures and/or tables, authored or reviewed drafts of the article, and approved the final draft.

- Youjia Zhang conceived and designed the experiments, analyzed the data, prepared figures and/or tables, authored or reviewed drafts of the article, and approved the final draft.
- Liang Zhang conceived and designed the experiments, performed the experiments, analyzed the data, prepared figures and/or tables, authored or reviewed drafts of the article, and approved the final draft.
- Zheng Gao conceived and designed the experiments, prepared figures and/or tables, authored or reviewed drafts of the article, and approved the final draft.
- Xiaolong Du conceived and designed the experiments, authored or reviewed drafts of the article, and approved the final draft.
- Feng Ran conceived and designed the experiments, prepared figures and/or tables, and approved the final draft.

### Human Ethics

The following information was supplied relating to ethical approvals (i.e., approving body and any reference numbers):

Nanjing Drum Tower Hospital, The Affiliated Hospital of Nanjing University Medical School.

### Data Availability

The raw data are available in the Supplemental File.

### Supplemental Information

Supplemental information for this article can be found online at http://dx.doi.org/10.7717/peerj.18189#supplemental-information.

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
