# Peer review of "Study on mid-term outcomes of atherectomy for patients with femoral popliteal artery lesions with different Global Limb Anatomic Staging System grades"

_PeerJ, doi:10.7717/peerj.18189_

## Round 0.1 · original submission · Major Revisions

Please follow the reviewers' requests strictly before re-submitting the manuscript. If the requests are not fulfilled, the manuscript may be rejected.

Reviewer 1 ·

Basic reporting

The structural organization of the text appears to lack coherence and meticulous attention, both in terms of formal presentation and linguistic refinement, thereby conveying an impression of insufficient dedication or effort invested in its development. Moreover, the statistical methodologies utilized remain inadequately elucidated. It is evident that the authors possess a substantial dataset, which, if appropriately harnessed by proficient individuals, holds promise for the generation of an insightful manuscript

Experimental design

The structural organization of the text appears to lack coherence and meticulous attention, both in terms of formal presentation and linguistic refinement, thereby conveying an impression of insufficient dedication or effort invested in its development. Moreover, the statistical methodologies utilized remain inadequately elucidated. It is evident that the authors possess a substantial dataset, which, if appropriately harnessed by proficient individuals, holds promise for the generation of an insightful manuscript

Validity of the findings

The structural organization of the text appears to lack coherence and meticulous attention, both in terms of formal presentation and linguistic refinement, thereby conveying an impression of insufficient dedication or effort invested in its development. Moreover, the statistical methodologies utilized remain inadequately elucidated. It is evident that the authors possess a substantial dataset, which, if appropriately harnessed by proficient individuals, holds promise for the generation of an insightful manuscript

Additional comments

1) Add the explanation of the acronym GLASS to your title and abstract. The first use of acronyms signifies the time of their explanation (global limb anatomic staging system)

2) “You can clearly feel the pulse of the distal artery and the filling of the veins” This form of casual language does not belong in a scientific manuscript.

3) As is evident from you abstract there is no comparison going on here. Please remove the term comparative from the title of the manuscript since this is an assessment of the device relevant to the GLASS stages.

4) 24 months is at best mid-term data and not long-term! Please rephrase

5) There is RESULT section missing from you abstract. Please be careful when submitting a manuscript since these are very important aspects which showcase a lack of commitment.

6) There needs to be an INTRODUCTION header

7) In you introduction you should explain what GLASS !!

8) Practicability? This is not a valid term in medicine

9) You should present a table with baseline study characteristics

10) It is GLASS stage and not phase

11) Line 87 Conclusions are presented at the end and not in the middle of the manuscript

12) You cannot asses vessel patency through a telephone conversation!!

13) Lines 173-78 this part does not belong in the definitions section

14) paired t-tests are used for intergroup comparisons ?? Where all your data normally distributed? How di you check for normality? These important parameters you have to explicitly describe !

15) Your results Section, should be subsectioned into the endpoints of intereset and not presented in a continuous text formal where you also discuss the outcomes, since there is the discussion section

·

Basic reporting

Basic reporting

The English diction and phraseology in this manuscript are the major weaknesses, which detract from the overall value of the paper. The English used is non-standard, with innumerable punctuation errors, and several unintelligible sentences. Some examples are:

Line 1-3: The title is missing one word: “ … patency rate of a peripheral plaque excision system…

A follow-up period of up to 24 months is not "long-term"; it is medium-term. The title should be corrected from "long-term patency" to 'medium-term patency'.

‘GLASS’ is not spelled out in the Key Words section or in the text until Line 218 for the first time, after the acronym GLASS has been used multiple times prior to Line 218; please spell out GLASS the very first time it appears in the text.

Line 51: “… SFA superfificial femoral artery… “; correctly: superficial

Line 93: "… stenosis of the femoral popliteal artery…"; maybe better would be: … stenosis of the femoral and/or popliteal artery".

Line 94: "… popliteal or subgenular arteries…"; I understand what you mean by “subgenular”, but this is not a term commonly used in English medical parlance. May I suggest instead: '… popliteal or infrapopliteal arteries'’; or: ‘… popliteal and other arteries distal of it…'..

Line 95: What is “CTO lesions”? Please spell out CTO rather than expecting the reader to know every acronym that you yourself might use in your everyday practice. Not all the readers of this paper might be vascular surgeons.

Line 102: "… passes through the endometrium". Surely, you do not mean "endometrium"?!? The endometrium is the inner epithelial lining of the uterus. Please correct.

Line 145-146: "If there is no contraindication for statins, they should be taken at the end of life." I understand what you mean but the English is incorrect. "At the end of life" means statins should be taken just before dying. Instead, please put: If there is no contraindication for statins, they should be taken for life.

Line 157: “ … usually scheduled each 3-5 months… “; correctly: every 3-5 months.

190-192: "During the surgery, 5 patients experienced… etc.". This sentence is too long and unintelligible. Please break it up into more than one and clearly-worded sentence. It is easiest to comprehend sentences that are no more than 14-16 words in length.

Lines 197 and 204: These 2 sentences are identical [1 should be deleted], including an error in English: "All patients did not require…"; Instead: None of the patients required…

Line 227: "… revascularization of belowknee lesions". Instead: '… Revascularization of below-knee lesions'.

Line 237: " … 2 patients experiencing amputations…"; Instead: 2 patients requiring amputations.

Line 242: "… ASO has been increasing year by year". Instead: '… ASO has been increasing year after year’; or: … ‘every year'.

Lines 288-291: "We found that 46-67%…etc"; This entire sentence does not make any sense and should be re-worded.

Lines 283-344: This is one single paragraph that is 61 consecutive lines [!] in length [which corresponds to ~1/6 of the entire length of the manuscript of ~366 lines]. Please break it up into at least 3, and preferably more, paragraphs at points when there is a new line of thought or a new topic introduced.

Line 308: " … Hypertension, diabetes, smoking, drinking and renal insufficiency"; Instead: … smoking, excessive alcohol consumption and renal insufficiency. [‘Drinking’ could be any type of beverage, not just alcohol.]

Line 308-309: "… recovered poorly…"; Instead: showed limited improvement.

Line 310: "… In addition,…"; Instead: In contrast,…

Line 311: "GLASS Class I in GLASS classification…"; Instead: GLASS Class I increased significantly [delete "in GLASS classification"; it is repetitive and unnecessary].

Lines 314-316: "Therefore, we can conclude… etc.". The sentence does not make sense. It is not the "long-term efficacy of grade 1 patients…". Patients do not have ‘efficacy’; the plaque excision procedure may have efficacy. This entire sentence has to be re-worded.

Line 326 is a repetition of line 320: Please use better English to avoid unnecessary repetition.

Line 340-341: "… patients with diabetes …" as a sentence is incorrect in English; please consider: '… patients with diabetes … likely to have knee lesions.”

Lines 345-349: This sentence does not make any sense. Please re-word it; preferably break it up into several sentences rather than 1 sentence that is excessively long. The best length for most sentences to be optimally intelligible is ~14-16 words, with some flexibility allowed depending on the thought expressed in any one sentence.

Line 349: "The surgical process should…" Instead: 'The surgical procedure should…'.

Line 356: “ …restenosis after plaque rotation…”; I presume you mean 'plaque rotablation’?

Line 358: What is "DCB"? This acronym has not been spelled out in the text.

Line 370-371: "It is grateful for all authors help with the preparation of figures in this paper." This sentence does not make any sense. It is not clear what you mean; please re-word it. Maybe: 'The authors are grateful for the preparation of the figures in this paper.'

Lines 375: "FUND"; do you mean 'FUNDING'?

Punctuation:
A great many punctuation marks [period, comma, colon, semicolon, etc.] in the text lack a space after them, which is a typographical error [e.g. Line 237: "… respectively.Feiring…", [correctly: … respectively. Feiring…; or Line 180: "IBM Corp,Armonk,NY,USA”; correctly: IBM Corp, Armonk, NY, USA; or all the reference numbers and the first authors of those references [e.g. [1]Cheng SWK, etc.
There is supposed to be a space between [1] and Cheng, or [2] and Hofling, etc.

In addition, the individual references should have an empty line between them rather than one continuous endless list of names and article titles.

The text of the manuscript seems to give the impression that the authors did not bother re-reading the manuscript but rather expected the reviewers to identify the innumerable errors in grammar. It would have been very easy to switch on the Spell Check app on the computer.

The whole manuscript should be re-written, preferably by someone else than the original author[s], since there seems to be a tendency to repeat the same errors. Bringing in a native English speaker or making use of professional translation services might be helpful.

Experimental design

Not applicable: This is not a research paper. Instead, it is a report on the same topic already published in earlier articles by other authors.

Intro and background: Provided. Satisfactory.

Structure conforms to PeerJ standards: The English language of the entire manuscript needs to be reviewed for linguistic correctness before it could be considered for publication.

Figures are relevant, high quality, well labeled and described: They are not of the highest quality, but they are acceptable.

Raw data supplied: Yes.

Original primary research: N/A. This is a report on the same topic already published in earlier articles by other authors. However, it has merit in that it adds to the world literature on this topic.

Fills in identified knowledge gap: N/A [not a research paper].

Rigorous investigation: N/A [not a research paper].

Methods can be replicated: N/A [the surgical procedures described are standard world-wide].

Impact and novelty: N/A [not original research].

Underlying data are statistically sound: Yes.

Conclusions are well stated: Yes. Provides indirect feedback on the efficacy of a surgical tool [TurboHawk peripheral plaque excision system] in patients with various degrees of disease severity.

Validity of the findings

The findings are clinically valid in that the paper provides indirect feedback on the efficacy of the surgical tool being used (TurboHawk peripheral plaque excision system).

Additional comments

References:
All individual references should have a break [empty] line between them rather than a long, continuous list of titles: [1]… Space, [2]… Space, [3]… Space, etc.

Please put a space between the reference number and the name of the author: [1] Cheng S WK; etc.

There are spelling errors in reference [1]: Endovascu lar: Endovascular; superfci al: superficial; Sur g: Surg; 133±140: 133-140.

Reference [10] is the same as reference [3]; please delete one of them.

Reference [40] is the same as reference [37]; please delete one of them.

Reference [44] is the same as reference [30]; please delete one of them.

Some of the references are quite old; e.g., [33] is from 1968 [56 years ago!]; [2] is from 1988 [36 years ago]. Maybe they are valid, but please try to focus on more recent articles.

The end of reference [21] is attached to the beginning of reference [22] in line 439; please re-arrange.

Line 446: “Schwarzwalder” has a typographical error; please correct.

Line 458: " …(outflflow lesion)…”; instead: ‘outflow lesion’.


Figures:

Figure 1 text: "… Patients with GLASS Class I in GLASS Classification increased…". Please delete 'in Glass Classification': It is obvious that ‘GLASS Class I' comes from the GLASS classification. It is unnecessary to repeat "in GLASS Classification". The same correction is needed in the text for Figure 3, Figure 4 and Figure 5.

In the text of Figures 5, 6, 7, 8, 9 and 10, please put a space between “GLASS II (III, IV)" and the word "classification" that follows: ‘GLASS II classification’, ‘GLASS III classification’, etc.

In the text of Figure 6, the sentence "The same with figure 4 []1[]" does not make sense; please correct or delete.

In the text of Figures 7, 8, 9 and 10, please make 2 separate sentences: The postoperative … different. There is no significant…

In the text of Figure 12: Box plot of the relationship between the right or left…: Instead '… right and left …’

In the text of Figures 13 and 14: "… after surgery on one side of the affected limb…", Please delete "… one side of…’
Also in Figures 13 and 14: "As one ages, the changes in ABI values…" is not professional or scientific English; instead please put: 'With advancing age, the ABI values will… '


Tables:

Table 1 text, at the end: "… before surgery recovered poorly"; instead: 'Before surgery improved marginally’

Table 1: 'Age [years]' and 'Sex [%]'.


Conclusion:

The English of the entire article is substandard and suffers from non-professional/non-medical phraseology. I recommend having the entire paper reviewed by a native English-speaker who is also familiar with medical terminology. Unfortunately, the suboptimal English detracts from the value of the manuscript to the point that it is not publishable in its current format, without corrections.

Recommendation:
Not recommended for publication until and unless the major flaws in English have been corrected. In its current format the manuscript is not good for the authors and not good for PeerJ.

---

## Round 0.2 · Minor Revisions

Please revise your manuscript according to the final requests of the reviewers.

·

Basic reporting

It is obvious that a great deal of effort has gone into correcting the original manuscript. In fact, in the great majority of the revised version the English is excellent. A few minor revisions remain, as follows.

Line 58: Please spell out CTA.

Line 122: ...116 out of 141...

Line 129-130: The sentence "The results show…" needs to be re-worded for greater clarity.

Line 151: ... highlighted …

Line 171: … a significantly longer …

Line 190: … contact of the drug with the endothelium …

Line 201: … evaluate the ...

Please introduce an empty line (line space) after Line 193 and after Line 251, just like in the rest of the article when starting a new paragraph with a new header.

Line 254-255: The sentence "It is grateful for all authors help with the preparation of figures in this paper" is unclear in English. Please revise.

Line 267: Figure 1. Pre- and post-intervention…

There should be an empty line (line space) separating the references from each other.

Table 1: Please put an empty space (letter space) between "Age' and "(years)" and also between "Sex" and "(%)".

A few punctuation errors remain throughout the text, usually the lack of a space after periods (.), e.g. Lines 40, 41, 44, 47, 49, 155, etc., but those typos do not detract from the contents of the paper.

Experimental design

Acceptable in the revised version.

Validity of the findings

Acceptable in the revised version.

Additional comments

The revised version of the manuscript is much improved, and the English is excellent in the great majority of the corrected manuscript. I recommend acceptance for publication after minor revisions as outlined above under "Basic reporting". Congratulations to the authors for reporting on a very important topic.

---

## Round 0.3 · accepted · Accept

Thank you for addressing the requests and suggestions of both reviewers.

·

Basic reporting

The corrected version of the manuscript displays use of the English language that is [almost] perfect. Rare and minor punctuation irregularities remain, but these do not detract from the scientific merit of the paper.

Experimental design

Suitable for publication following the corrections.

Validity of the findings

Suitable for publication following the corrections.

Additional comments

Congratulations to the authors on this most valuable contribution to the world literature.